# Quantification of SPECT Concentric Ring Artifacts by Radiomics and Radial Features

**Emilio Mezzenga** [1], **Anna Sarnelli** [1,*], **Giovanni Bellomo** [2], **Frank P. DiFilippo** [3], **Christopher J. Palestro** [4] **and Kenneth J. Nichols** [4]

1 Medical Physics Unit, IRCCS Istituto Romagnolo per lo Studio dei Tumori (IRST) "Dino Amadori", 47014 Meldola, Italy; emilio.mezzenga@irst.emr.it
2 Neurology Clinic, Department of Medicine and Surgery, University of Perugia, 06132 Perugia, Italy; giovanni.bellomo@unipg.it
3 Department of Nuclear Medicine, Cleveland Clinic, Cleveland, OH 44195, USA; difilif@ccf.org
4 Department of Radiology, Donald and Barbara Zucker School of Medicine at Hofstra/Northwell, Hempstead, NY 11549, USA; palestro@northwell.edu (C.J.P.); knichols@northwell.edu (K.J.N.)
* Correspondence: anna.sarnelli@irst.emr.it

**Abstract:** (1) Background: Concentric ring artifacts in reconstructed SPECT images indicate the presence of detector non-uniformity in gamma camera systems. The identification of these artifacts is generally visual and not quantitative. The aim of our study was to evaluate observer assessments of the presence of concentric rings in reconstructed SPECT phantom images and to verify whether quantitative texture analysis can detect such artifacts, which are detrimental to accurate tumor detection. (2) Methods: Test data were acquired as part of the quarterly quality assurance program using a standardized SPECT phantom containing solid spheres, solid rods, and a water solution of $^{99m}$Tc. Forty separate SPECT acquisitions were analyzed to assess the presence of ring artifacts. Two experienced medical physicists independently reviewed transaxial images and graded the severity of artifacts on a five-point scale. Quantitative radiomic features were computed for volumes of interest located in the uniform phantom section. In addition to these, radial contrast ($R_{Contrast}$) and radial root-mean-square contrast ($R_{RMSC}$) were also calculated and derived from the radial profile of summed slices transformed into polar coordinates. (3) Results: Artifacts were considered sufficiently severe to warrant camera re-tuning in 10 rod sections, 17 sphere sections, and 16 uniform sections. In the uniform sections, there was "good agreement" for inter-observer and intra-rater assessments ($\kappa$ = 0.66, Fisher exact $p < 0.0001$ and $\kappa$ = 0.61, and Fisher exact $p = 0.001$, respectively). The two radial features agreed significantly ($p < 0.001$) with visual severity judgment of ring artifacts in uniform sections and were selected as informative about the presence of ring artifacts by LASSO approach. The increased magnitude of $R_{Contrast}$ and $R_{RMSC}$ correlated significantly with increasingly severe artifact scores ($\rho$ = 0.65–0.66, $p < 0.0001$). (4) Conclusions: There was good agreement between the physicists with respect to the presence of circular ring artifacts in uniform sections of SPECT quality assurance scans, with the artifacts accurately detected by radial contrast and noise-to-signal ratio measurements.

**Keywords:** image artifacts; radiomics; SPECT system; gamma camera; quality assurance; image texture analysis; inter-observer agreement; phantom

## 1. Introduction

Numerous factors influence the accuracy of single-photon-emission computed tomography (SPECT) reconstructions. In addition to patient motion issues (e.g., respiratory and cardiac), complications include photon attenuation and scatter, the variability of point spread function with distance from detectors, imperfect detector efficiency, and low counting statistics [1]. Over the years, considerable effort has been expended to address these issues through the application of physics to optimize the reconstruction process, together

with electric and mechanical engineering to improve SPECT system designs. However, data acquisition problems can occur abruptly, or as a result of the gradual electrical or mechanical deterioration of SPECT system components, which can compromise the reliability of acquired data [2] to correctly evaluate tumor progression/regression.

For this reason, adherence to regularly scheduled standardized quality assurance (QA) procedures is important to identify problems with data acquisition before they can adversely affect clinical readings. These QA procedures include quarterly SPECT phantom reconstructions, which are typically assessed visually for contrast, spatial resolution, and uniformity of response [3].

In the course of evaluating SPECT phantom transaxial reconstructions to assess the integrity of acquired image data, the appearance of conspicuous concentric ring artifacts often triggersthe re-tuning of the detectors, as these artifacts may be due to inadequate uniformity corrections [4]. One cause of this can be the insufficient count density of the correcting flood field, the details of which have been studied extensively [5]. The choice of tomographic reconstruction filters affects the appearance of artifacts, and perceptibility of artifacts depends on their size and their position within the object [6].

However, these evaluations are visual, not quantitative, and there are no published criteria for determining what constitutes an artifact that is sufficiently severe to necessitate detectorre-tuning. When medical imaging physicists convene to read QA studies from other institutions, they begin with training data sets to calibrate their readings and to minimize the perceptual differences among them, and they read by consensus. However, factors such as brightness, contrast, and the screen resolution of the display monitor can influence visual impressions.

To address these issues, algorithms have been developed to analyze standardized SPECT QA phantoms, both to report the results of spatial resolution and contrast results [7,8], and to report the possibility of artifacts [9]. Some of these algorithms are based on texture analysis [7,9]. Approaches to quantifying the severity of circular ring artifacts include annular sampling techniques [10] and the use of the Hough transform, the efficacy of which for detecting artifacts has been assessed using computerized Monte Carlo phantom simulations [11].

Our study was conducted with two goals: (1) to determine how well medical physicists (who are experienced at performingthese evaluations) agree with one another about the presence and severity of concentric ring artifacts, and (2) to determine whether there are any quantitative image metrics that correspond to visual judgments regarding the significance of these artifacts. The use of a quantitative parameter could facilitate the decision as towhether corrective action needs to be undertaken.

## 2. Materials and Methods

### 2.1. Phantom Data Acquisitions

All the test data were acquired as part of routine quarterly QA using the Jaszczack phantom [3]. The phantom included a cylindrical water bath, with acrylic inserts of six rod sizes, solid spheres of six sizes, and a uniform volume devoid of rods or spheres [12,13]. Phantoms were loaded with a water solution containing 666–740 MBq $^{99m}$Tc. Phantoms were positioned on the patient imaging table, not in a specialized phantom holder. Each 360° set of projection data was comprised of 128 projections acquired as 128 × 128 matrices with 32–38 × $10^6$ total counts. Magnification factors of 1.00–1.85 were used (pixel size = 3.3 ± 0.7 mm; range = 2.1–5.2 mm).

All data sets were reconstructed by filtered back projection (FBP) with a Hanning post-filter (cutoff = 1.0 cm$^{-1}$)and corrected by a simple Chang attenuation correction with attenuation coefficient of $\mu = 0.11$ cm$^{-1}$, or adjusted as necessary in order to obtain the most uniform appearance of transaxial reconstructions of the volume of the phantom containing no rods or spheres.

From more than 200 acquisitions performed for 12 different SPECT systems between 2016–2020, a medical physicist (KJN) with over twenty years' experience in assessing SPECT system phantom testsidentified40 studies (Table 1) at the time of data acquisition as either

having evidence of concentric ring artifacts; others were acquired to determine whether concentric ring artifacts were resolved following camerare-tuning performed within 1 week after obtaining an unacceptably non-uniform-appearing result. A 41st phantom study, for which observers agreed that no artifacts were visible, was used as a reference standard. At the time of the QA phantom data acquisitions, camera ages ranged from 1 month to 21 years old, with a mean age of $13.2 \pm 6.2$ years (median 15.0 years). Ten systems were dual-detector systems and twowere single-detector systems. Considering the mean age of the SPECT systems that generated the data, most of the studies were generated by old scanners, which tend to drift electronically more frequently than newer machinery.

**Table 1.** SPECT systems and related number of studies selected for concentrinc ring artifact evaluation.

| Manufacturer | Model | Studies |
|---|---|---|
| General Electric | Millenium—dual detector | 17 |
| Philips | Skylight—dual detector | 10 |
| Philips | Adac Argus—single detector | 4 |
| Philips | Adac Argus—single detector | 3 |
| Siemens | Symbia Intevo—dual detector | 2 |
| Siemens | Symbia Intevo—dual detector | 2 |
| Siemens | Symbia Intevo—dual detector | 3 |

*2.2. Visual Phantom Readings*

Automated algorithms written in IDL 8.4 were run on all transaxial phantom data that generated a series of output jpg image files [7]. Because some accrediting agencies request displays of all transaxial slices [3], multiple jpg files were generated for each phantom for all transaxial slices spanning the entire height of the phantom, and a separate composite jpg file was generated, showing a composite of threeimages (Figure 1): a summed rod section, asingle section through spheres with the highest contrast, and a section of uniform activity concentration. The jpg files were reviewed and scored by two observers for the presence and severity of concentric ring artifacts.

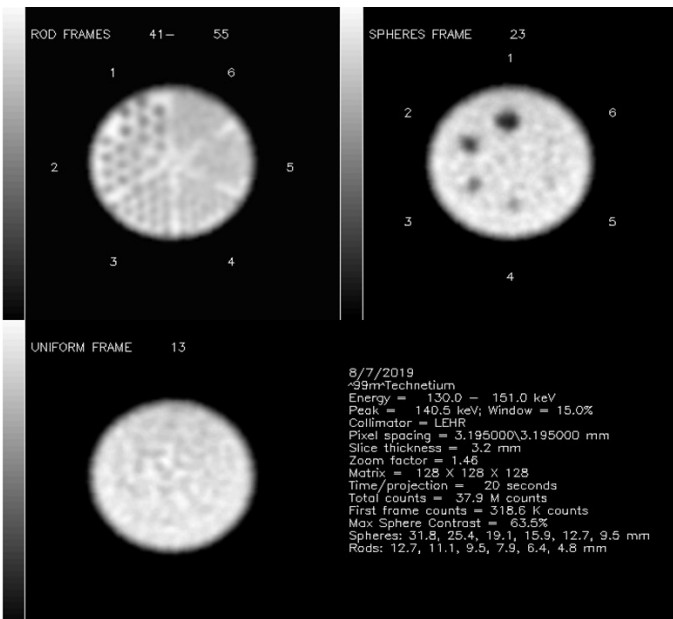

**Figure 1.** Example of composite image. In this case, both readers agreed there were no perceptible artifacts in any phantom section.

To gauge inter-rater agreement about the presence and severity of concentric artifacts, two experienced medical physicists (KJN and FPD), each with over twenty years'

experience in assessing SPECT system phantom tests, independently scored severity of artifacts on a five-point scale (0 = "no artifact"; 1 = "probably insignificant artifact"; 2 = "equivocal"; 3 = "probably significant artifact"; 4= "severe artifact definitely requiring camerare-tuning"). Singlescores were generated by each reader for the perception of artifacts as seen on the composite summary images (Figure 1), and these were defined separately for rod sections (Figure 2), sphere sections (Figure 3), and uniform phantom sections (Figure 4).

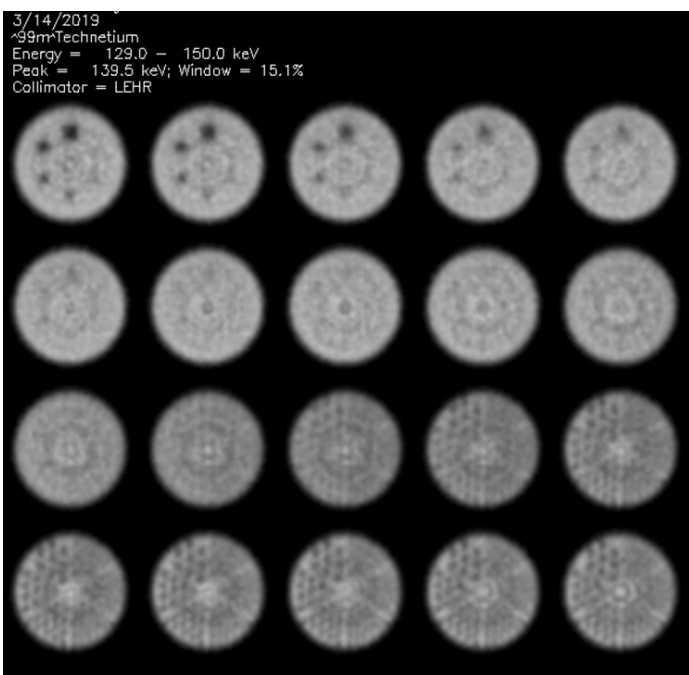

**Figure 2.** Example of images related to the spheres sections and rod sections of a phantom, whichboth readers scored "4", forsevere concentric ring artifacts, in both sections.

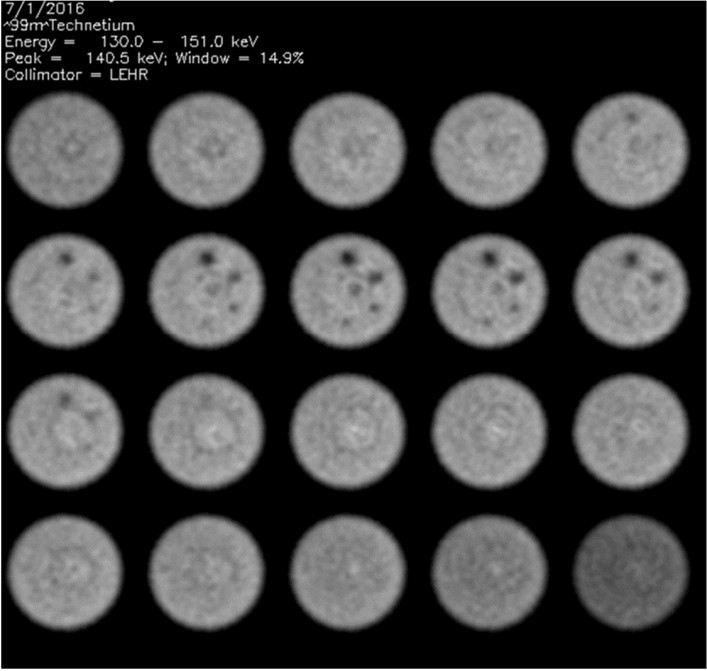

**Figure 3.** Example of images related to the uniform sections and spheres sections of a phantom, which both readers scored"3", for probably significant concentric ring artifacts, in both sections.

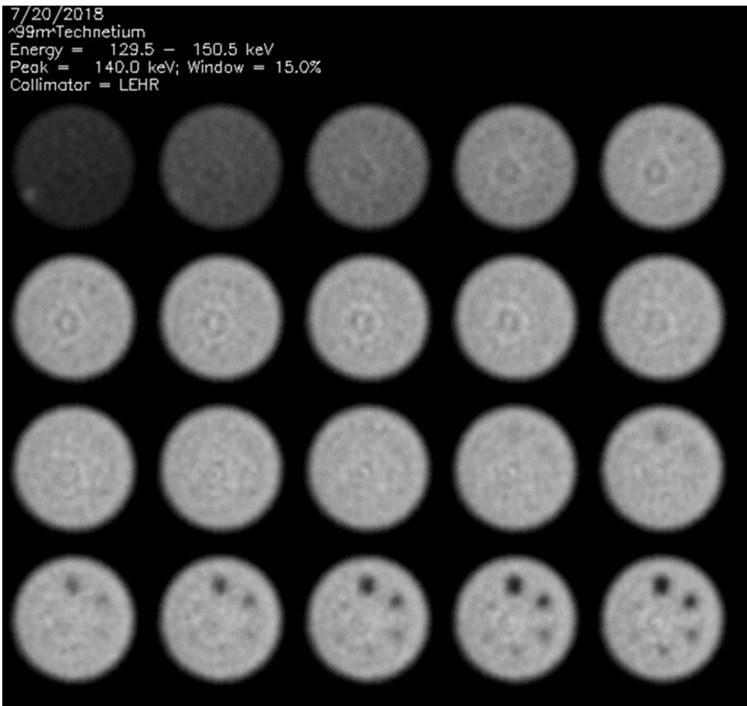

**Figure 4.** Example of images related to the uniform sections and spheres sections of a phantom, which both readers scored "4", forsevere concentric ring artifacts, in both sections.

One physicist (KJN) re-scored the images blinded to his initial scores and to those of the other reader, to enable the assessment of intra-rater reproducibility. Readers had no knowledge of any texture analysis metrics values; these values were not computed until after all readings were completed by both readers.

To compute cut-off values for the presence of ring artifacts, scale ratings were dichotomized so that any reading >2 was assigned 1 and any reading <2 to 0. Ambiguous readingsequal to 2 were excluded from this analysis.

### 2.3. Radiomic Features Extraction

Texture analysis was performed by two medical physicists (EM and AS) on DICOM files of transaxial reconstructions. SPECT image data were imported to MIM software v7 (MIM Software, Inc., Cleveland, OH, USA) [14], where cylindrical volumes of interest (VOIs) were drawn on uniform SPECT sections.VOIs were centered on the middle of a transaxial section individually for each phantom data set. Finally, each dataset, together with its own VOI, was processed by Image Biomarker Standardization Initiative (IBSI) compliant [15] SIBEX software [16]. Following the suggestions of the IBSI before computing features, a pre-processing step was implemented: fixed bin number of 32 and alignment of grid centers were used for intensity discretization and for the interpolation grid, respectively. A total of 163 three-dimensional image texture features were computed (Table 2) and reported in Table S1 of Supplementary Materials.

**Table 2.** SIBEX software radiomics family name and their IBSI identification [16,17]. 3D = three-dimensional; mrg = merged. The number in rounded brackets refers to the total number of features considered in that family.

| Family Name | Identification |
|---|---|
| Gray-Level Co-Occurrence Matrix (25) | GLCM (3D:mrg) |
| Gray-Level Distance Zone Matrix (16) | GLDZM (3D) |
| Gray-Level Run-Length Matrix (16) | GLRLM (3D:mrg) |
| Gray-Level Size Zone Matrix (16) | GLSZM (3D) |
| Neighborhood Gray-Level Dependence Matrix (17) | NGLDM (3D) |
| Neighborhood Gray-Tone Dependence Matrix (5) | NGTDM (3D) |
| Intensity-Based Statistics (18) | IS |
| Intensity Histogram (23) | IH |
| Intensity Volume Histogram (7) | IVH |
| Morphological (20) | MORPH |

### 2.4. Radial Features Extraction

Considering the radial nature of concentric ring artifacts, two more image features were computed: radial contrast ($R_{Contrast}$) and radial root-mean-square contrast ($R_{RMSC}$) [17]. Custom macros were written for ImageJ software to compute $R_{Contrast}$R and $R_{RMSC}$R from the 40 SPECT phantom images $f(x, y, z)$ [18]. The workflow used to obtain radial features is illustrated in Figure 5. Depending on voxel dimension, 8–10 tomographic homogeneous phantom transaxial sections ($z_H$) were summed together:

$$f_z(x, y) = \sum_{z_H} f(x, y, z) \tag{1}$$

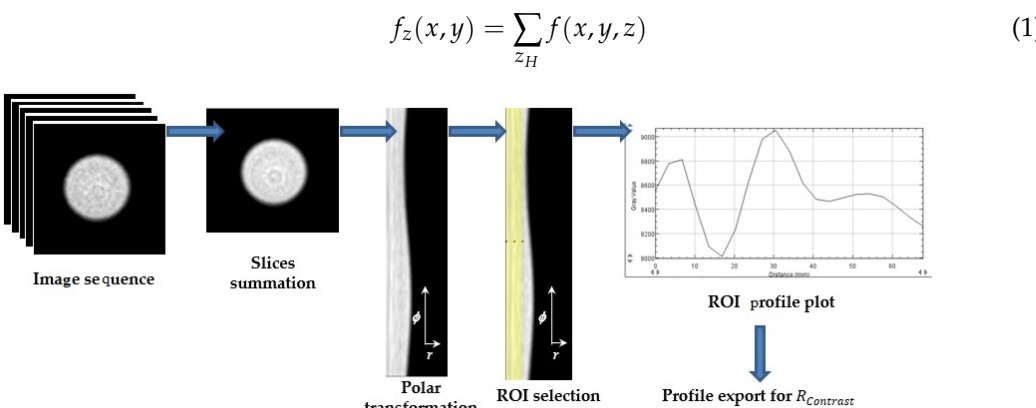

**Figure 5.** Workflow used to obtain radial features (i.e., $R_{Contrast}$ and $R_{RMSC}$) from SPECT data sets.

Summed images $f_z(x, y)$ were transformed to polar coordinates $f_z(r, \phi)$, where:

$$r = \sqrt{x^2 + y^2} \tag{2}$$

and:

$$\phi = \tan^{-1}\left(\frac{y}{x}\right) \tag{3}$$

To obtain the radial profile $f_{z\phi}(r)$, the images $f_z(r, \phi)$ were summed over the $\phi$ angular variable:

$$f_{z\phi}(r) = \sum_{\phi} f_z(r, \phi) \tag{4}$$

This profile was used to compute $R_{Contrast}$ and $R_{RMSC}$ as follows:

$$R_{Contrast} = \frac{Max\left(f_{z\phi}(r)\right) - Min\left(f_{z\phi}(r)\right)}{Max\left(f_{z\phi}(r)\right) + Min\left(f_{z\phi}(r)\right)} \tag{5}$$

$$R_{RMSC} = \frac{\sigma(f_{z\phi}(r))}{\mu(f_{z\phi}(r))} = \frac{\sqrt{\frac{1}{K}\sum_k \left(f_{z\phi}(r_k) - F\right)^2}}{\frac{1}{K}\sum_k f_{z\phi}(r_k)} \tag{6}$$

with $\mu$ and $\sigma$ being the estimates of the mean and standard deviation of the radial profile $f_z(r, \phi)$, respectively, $k$ denotes the spatial location and $K$ the total number of spatial locations.

*2.5. Statistical Analysis*

Statistical analysis was performed using MedCalc software [19]. Trends of variables with increasing severity of concentric ring artifacts were quantified by $\chi^2$ rank correlation with Spearman's ρ on all the 40 SPECT images. The discriminative power of radiomic and radial features was investigated by means of the least absolute shrinkage and selection operator (LASSO) approach [20]. The following nested cross-validation procedure was adopted: the whole data set was randomly split into a training set (70%) and a test set (30%).The LASSO shrinkage penalty parameter was tuned on training data through a fivefold cross validation, selecting the value with one standard deviation of the minimum error, and the resulting logistic model was evaluated on the test set [21]. The process was repeated 30 times and the most frequently selected features across iterations were reported as the most informative features. To estimate LASSO-logistic model performance on radiomic and radial features, accuracy, identified as the mean area under the receiver operating curve (AUC) on the test sets across iterations, was calculated. Ambiguously classified phantom images were classified by each iteration using the Youden index [22] as the threshold, and the most frequent classification over iterations was used to estimate the presence of ring artifacts. The whole procedure was performed by using the R glmnet package and the R pROC package (R software v4.0.4).

Correlation between visual scores and radiomic and radial features was evaluated by Spearman's coefficient (ρ) of rank correlation.

Inter-rater agreement and intra-rater reproducibility of visual scores was determined by the kappa (κ) statistic, for which strength of agreement is considered "poor" for κ < 0.20, "fair" for κ = 0.21–0.40, "moderate" for κ = 0.41–0.60, "good" for κ = 0.61–0.80, and "very good" for κ ≥ 0.81 [23], with significance determined by Fisher's exact test.

All tests were two-sided and a probability-value (*p*) < 0.05 was considered to be statistically significant.

## 3. Results

*3.1. Visual Scores*

The count levels were similar among the phantom acquisitions, with $(34 \pm 1.0) \times 10^6$ counts, and were not normally distributed (kurtosis = 6.0, $p = 0.0004$), as they were narrowly clustered near the mean. Consequently, differences in counts or innoise-to-signal ratios among phantoms were unlikely to account for the perceived artifacts.

Among the 40 phantom acquisitions, artifacts were considered sufficiently severe, based on the visual scores, to require camerare-tuning for 10 rods sections, 16 sphere sections, and 21 uniform sections for the first reader and for 10 rods sections, 17 sphere sections, and 16 uniform sections for the second reader. Thus, the percentage of phantoms in which artifacts were perceived wassmaller in the rod than in the uniform sections (24% versus 39–51%, $p = 0.01$). From the displays of all individual transaxial sections, inter-observer comparisons demonstrated "fair agreement" as to artifacts for rods (κ = 0.33, $p = 0.09$), "good agreement" for spheres (κ = 0.64, $p < 0.0001$), and "good agreement" for uniform sections (κ = 0.65, $p < 0.0001$). The correlation between readers was stronger for the sphere and uniform sections than for the rod sections ($p = 0.004$).

Intra-observer reproducibility demonstrated "moderate agreement" for rods (κ = 0.58, $p = 0.0001$), "good agreement" for spheres (κ = 0.70, $p < 0.0001$), and "good agreement" for uniform sections (κ = 0.70, $p < 0.0001$).

For the perception of the artifacts seen on the composite images, there was "moderate agreement" for inter-observer comparisons ($\kappa = 0.47$, Fisher exact $p = 0.01$) and "good agreement" for intra-observer reproducibility ($\kappa = 0.68$, $p < 0.0001$).

### 3.2. Image Analysis Metrics

The LASSO approach indicated that two radial features, $R_{Contrast}$ and $R_{RMSC}$, had high predictive performance, with 20 and 13 occurrences over the 30 iterations, respectively. $R_{Contrast}$ and $R_{RMSC}$ also showed significant rank correlation with the visual scores, with $\rho = 0.66$ ($p < 0.001$) and $\rho = 0.65$ ($p < 0.001$), respectively. The LASSO logistic model's performance in terms of mean AUC (an estimate of accuracy) was 80% (standard deviation = 16%). Concerning the "equivocal" readings (visual score = 2), two remained ambiguous, with 50% of iterations detecting the presence of ring artifacts, while the other 50% did not detect them. For the remaining artifacts, most of the iterations (>80%) agreed with the presence or absence of rings.

## 4. Discussion

As with any type of device used to collect data that contribute to forming a clinical diagnosis, it is important to assess the performance characteristics of rotating SPECT systems. SPECT equipment testing includes the acquisition and analysis of three-dimensional phantoms to gauge system tomographic spatial resolution, image contrast, and uniformity of response; it is recommended to be performed as part of acceptance testing [24], and thereafter on a quarterly basis [3]. The failure to recognize concentric ring artifacts can contribute to the misinterpretation of nuclear cardiology studies [25], and non-cardiac nuclear medicine studies [5].

Determining whethera particular image feature is visible is not trivial. One of the goals of our investigation was to document the extent to which two independent medical imaging physicists agreed on the appearance and severity of artifacts. The respective thresholds of visible detection in a $34 \times 10^6$ count phantom study and in a clinical study likely differ. Observer experience is a factor, and evaluations, even by experienced physicists, might differ. Visibility indexes, such as those dependent on feature contrast and feature dimensions, can help in this regard [26], but ultimately such indices require calibration with scores assigned by human observers [27].

We found that readers agreed more frequently when observing all of the transaxial sections rather than viewing only the summary images.Our readers had "good agreement" whendetecting artifacts in uniform and sphere sections but only "fair agreement" about rods sections. The $\kappa$ values of reader agreement and reading reproducibility were higher for the uniform sections and for the sphere sections than for the rod sections. This may have been due to the fact that there was less distraction in the uniform sections in perceiving ring artifact patterns compared to the rod sectors, for which the rod patterns were superimposed on count variations within and between rod images of different sizes.

Besides deciding whether a concentric ring artifact is visible, deciding whether an artifact is sufficiently severe to cause potential clinical reading problems is another matter, because this requires removing a camera from clinical service for long enough to performa thorough re-tuning of the detectors. The process of camera re-tuning may be fairly unobtrusive, as in routine detector tuning or flood calibration by a technologist of $100–120 \times 10^6$ counts; or it can be more time-consuming, involving a full calibration of energy, linearity, and uniformity corrections by a field service engineer. Consequently, the use ofa quantitative basis for triggering the decision to retune detectors would offer useful information for technologists performing quarterly SPECT quality assurance scans.

While the results of quality assurance phantom scans are usually assessed visually, recent progress has been reported on quantifying scanner performance automatically [7–9,11]. Among the parameters that are evaluated in SPECT phantom scans, non-uniformity is perhaps the most challenging. O'Connor et al. [6] found that the choice of reconstruction filters has a pronounced effect on the manifestation of planar flood field non-uniformity

and on the formation of tomographic concentric ring artifacts, and that the ability of observers to perceive such artifacts depends on the size and location of these artifacts; smaller artifacts closer to the center-of-rotation were less perceptible than larger ones further from the center [6]. This is consistent with a reader tending to give more credence to a pattern that is perceived to be repeated over more pixels than over fewer pixels, as there would be a tendency to attributemore weight to a similar pattern confirmed in multiple locations, even when these deviations from neighboring counts are small. Concentric ring artifacts constitute this type of pattern, and, as such, the greater the radius, the more likely a human observer would be todetect it, even if it is subtle.

An advantage of the polar coordinate transformations used in our study, and in the application of the Hough transform implemented by Hirtl et al. [11], is the fact that non-uniformities tend to be concentric rings, so that ring artifacts of any size become mapped onto straight lines across the entire extent of the image matrix in polar coordinates. Polar coordinate transformations have the effect of bringing concentric artifacts that are near to the center-of-rotation up to an equal footing with those at a greater radius. This process depends on knowing the center of the reconstruction matrix, which generally corresponds to the center field of view in each slice. This implies that in order to compute radial features, the uniform section of the cylindrical phantom should be placed as close as possible to the center of rotation of the detector. In practice, the presence of a small offset between the phantom and rotation axes does not constitute a problem as long as this offset makes it possible toextract a radial profile with a sufficient number of pixels, which, from our experience, corresponds to an offset smaller than half of the phantom radius.

The fact that the $R_{Contrast}$ and $R_{RMSC}$ parameters agreed more closely with our medical physicists' scores than any of the 163 radiomic features can likely be explained by the fact that the radial parameters make use of the a priori condition that these artifacts are circular patterns, whereas the radiomics features are more generic and shape-independent.

*Limitations*

To maximize spatial resolution, patient data are usually acquired with contouring instead of with circular orbits. Our acquisitions were performed with contouring, but considering that the phantom we used was a simple cylinder, these orbits were likely close to being circular. It has been shown that non-uniform planar flood field corrections create circular artifacts in SPECT transaxial reconstructions for orbits witha fixed radius, but more complicated shapes when elliptical orbits are used [5].

With respect to thereconstruction process, in this study, the SPECT phantom images were reconstructed by means of FBP. This reconstruction was the one originally specified by the American College of Radiology (ACR), and the filter that ACR now recommends produces reconstructions that are quite similar. Moreover, the NEMA NU-1 2018 publication states that: "The prescribed image evaluation steps for reconstructed resolution, which dictate filtered back-projection, were chosen as appropriate to the standard Gamma camera design. This technique may not be appropriate for novel SPECT systems that employ advanced reconstruction algorithms and where filtered back-projection is not the preferred or appropriate reconstruction technique." Since our SPECT systems (Table 1) have a median age of 15years, we decided tostandardize image reconstruction by using FBP reconstruction for all SPECT systemsindependentof system age.Further investigations could be useful to determine whether our findings are affected by using OSEM reconstruction.

Our study was limited to documenting the degree to which medical physicists agree with one another in their perception of the presence and severity of artifacts, not over-whether there were problems in the actual clinical studies. By linking $R_{Contrast}$ and $R_{RMSC}$ to their visual impressions, we have provided a means to connect quantified imaging parameters computed from standardized phantom scans to determine whether an artifact would have been judged to be sufficiently serious to require service on a SPECT system. We must, however, point out that the whole analysis conducted to identify the most informative features was really proof of principle due to the small sample size. The goal

was to assess whether radial or radiomic features were able to discriminate between the presence or absence of ring artifacts. Alarger dataset and a combination of feature selection techniques would undoubtedly provide more robust results.

Nonetheless, the results concerning the most discriminative features serve as a guideline for acting on the results of phantom scans. There are, in fact, innumerable variables involved in forming a final clinical diagnosis.

Our analyses of radiomic features and radial parameters were performed only in the uniform sections, not in the sphere or rod transaxial sections. The inter-rater and intra-rater agreement measures were lowest for the rod sections, indicating that it was more difficult to agree on the presence and severity of artifacts in those sections than in the sphere and uniform phantom sections. However, we found that the concentric ring artifacts were detected by both readers throughout the individual phantom sections. The fact that some radiomic features agree well with visual impressions in uniform sections suggests that further work is warranted to enable texture analysis applications to non-uniform phantom areas containing spheres and rods, where artifacts also appear.

The values of textural radiomic features can be affected by acquisition and reconstruction parameters, as reported for CT and PET imaging [28–30], and these influences require further investigation for SPECT imaging.

## 5. Conclusions

There was good agreement among observers, with reproducible results, as to the presence of circular ring artifacts in uniform sections of SPECT quality assurance scans, and some texture analysis metrics agreed well with visual impressions in uniform sections. Among all the computed image metrics, $R_{Contrast}$ and $R_{RMSC}$, derived from the radial profiles of summed slices transformed into polar coordinates werethe most effective metrics for identifying severe concentric ring artifacts in uniform sections of test phantom data, and they were the only metricsthat significantly correlated with the visual scores ($\rho = 0.66$ and $\rho = 0.65$, respectively). These quantified variables thus have the potential to support subjective visual evaluation, therebypromoting the automation of SPECT quality assurance and aiding the assessment of the degree to which efforts are successful in remedying detected equipment deficiencies.

**Supplementary Materials:** The following are available online at https://www.mdpi.com/article/10.3390/app12052726/s1. Table S1: All features considered for each family name feature.

**Author Contributions:** Conceptualization, E.M., A.S. and K.J.N.; methodology, A.S., K.J.N., F.P.D., G.B. and E.M.; software, K.J.N., E.M. and A.S.; validation, A.S., K.J.N., F.P.D., G.B. and E.M.; formal analysis, E.M., A.S., G.B. and K.J.N.; investigation, A.S., K.J.N., E.M., F.P.D. and G.B.; resources, A.S.; data curation, A.S.; writing—original draft preparation, E.M., A.S., C.J.P. and K.J.N.; writing—review and editing, G.B., C.J.P., K.J.N. and F.P.D.; supervision, A.S. and K.J.N. All authors have read and agreed to the published version of the manuscript.

**Funding:** This research received no external funding.

**Institutional Review Board Statement:** Not applicable.

**Informed Consent Statement:** Not applicable.

**Acknowledgments:** The authors thank Fritzgerald Leveque, May Liu, and Christopher Caravella for their invaluable assistance in collecting and processing the phantom data analyzed in this investigation.

**Conflicts of Interest:** The authors declare no conflict of interest.

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
