# Peer review of "Quantification of SPECT Concentric Ring Artifacts by Radiomics and Radial Features"

_applsci, doi:10.3390/app12052726_

Round 1
Reviewer 1 Report
This manuscript is well organized and written. The results can have a practical value.
My suggestion is to take a closer look on definition of the RNSR feature. It is called "noise-to-signal ratio"; however, the fluctuations that contribute to the numerator of (1) come mainly from the deterministic signal (which is the radial oscillation representing the artefacts of interest, Fig. 5). The noise, which is attributed to a random component of fzΦ(r) is of much lower magnitude, and is negligible in Fig. 5, at least visually.
Even if the radial profile be a random noise, one does not know its mean μ and standard deviation σ. These quantities could only be estimated in the described situation. Then, one can talk about the estimates of μ and σ.
If (1) is reformulated, such that s = sqrt[K-1∑Kk=1(fzΦ(rk)-F)2], which is a root-mean-square of the profile signal, is placed in the numerator and the average F = ∑Kk=1fzΦ(rk) in the denominator, the resulting coefficient can be called "root-mean-square contrast".
Author Response
"Please see the attachment."

Reviewer 2 Report
This study evaluated the consistency of two experienced medical physicists in identifying and rating the severity of concentric ring artefacts on SPECT phantom images obtained as part of routine QA exercise on multiple SPECT cameras of varying ages. Subsequently, the authors performed a textural analysis of the DICOM images of the transaxial reconstructions. In addition, two radial features, radial contrast and radial noise-to-signal ratio, were calculated. The authors reported a good inter-rater agreement in identifying concentric ring artefact. The two radial features agreed significantly with visual assessment.
The study evaluated a subject of potential clinical importance. Quantification of the severity of concentric ring artifact seen on phantom images acquired as part of camera QA may be necessary to differentiate artefacts severe enough to impact the interpretation of clinical images and those that are not.
Specific comments
- Methods: “…40 studies were identified at the time of data acquisition as either having evidence of concentric ring artifacts, or else were acquired to determine whether concentric ring artifacts were resolved following camera retuning performed within 1 week after obtaining an unacceptably non-uniform-appearing result.” Were the two medical physicists who rated the images used in this study among the physicists who read the scan as part of routine QA? This clarification is important to show potential readers the inter versus intra-individual consistency in the identification of concentric ring artifacts in phantom QA images.
- Methods: Line 102 – Please include the model and manufacturer information of the SPECT cameras here.
- Methods: Lines 126-127 - Please quantify the experience of the two medical physicists in terms of years each has been interpreting this kind of QA scan.
- Please be consistent in the way you write re-tuning. I think “re-tuning” may be preferable to “retuning”. Please correct throughout the manuscript.
- With improvements in the speed of modern cameras, many users now perform reconstruction using the iterative algorithm. Do the authors think their choice of filtered back projection could have influenced the prevalence and severity of concentric ring artifacts? Please include a thought on this in the discussion.
- The ages of the SPECT cameras on which the phantom images were acquired range from a month to 20 years. Please include the mean and SD or median and range of the ages of the camera to give readers a feel of how old the cameras are. In addition, do the authors think that the ages of the cameras (some of the cameras appear to be quite old) could have influenced the quality of the phantom images?
- It is surprising that the authors decided to include only 40 images that have been previously assessed to have concentric ring artifacts in the first part of the study assessing inter-rater agreement in identifying the presence of this artifact. This represents a selection bias as the readers are biased into calling concentric ring artifact. I think it would be more robust to include all phantom images acquired during the study. This would have allowed for testing of the readers' consistency in identifying normal scans without concentric ring artifact as well as abnormal scans with concentric ring artifacts and their severity.
Author Response
"Please see the attachment."
